# The Impact of Vegan Diets on Indicators of Health in Dogs and Cats: A Systematic Review

**DOI:** 10.3390/vetsci10010052

**Published:** 2023-01-12

**Authors:** Adriana Domínguez-Oliva, Daniel Mota-Rojas, Ines Semendric, Alexandra L. Whittaker

**Affiliations:** 1Neurophysiology, Behavior and Animal Welfare Assessment, DPAA, Universidad Autónoma Metropolitana (UAM), Mexico City 04960, Mexico; 2School of Biomedicine, The University of Adelaide, Adelaide, SA 5005, Australia; 3School of Animal and Veterinary Sciences, Roseworthy Campus, The University of Adelaide, Roseworthy, SA 5116, Australia

**Keywords:** plant-based diets, animal health, vegan companion, animal diets

## Abstract

**Simple Summary:**

There has been controversy within the scientific literature, and in the popular press and online media, around the safety of feeding vegan diets to dogs and cats. With an increase in adherence to meat-free diets in the human population, many guardians may be considering providing these diets to their companion animals. Concerns arise due to dog and cat gut physiology which has adapted to a complete meat-based diet (cats) or largely meat-based diet (dogs). Particular concerns have been raised around deficiencies in certain amino acids such as taurine, and vitamins such as B12 (cobalamin) and B9 (folate). To date, there has been no formal assimilation of the scientific evidence on this topic, with a focus on actual health impacts of diets, as opposed to nutritional composition. In this review, we conducted a formal assessment of the evidence in the form of a systematic review. We found that there has been limited scientific study on the impact of vegan diets on cat and dog health. In addition, the studies that have been conducted tended to employ small sample sizes, with study designs which are considered less reliable in evidence-based practice. Whilst there have been several survey studies with larger sample sizes, these types of studies can be subject to selection bias based on the disposition of the respondents towards alternative diets, or since answers may relate to subjective concepts such as body condition. However, there is little evidence of adverse effects arising in dogs and cats on vegan diets. In addition, some of the evidence on adverse health impacts is contradicted in other studies. Additionally, there is some evidence of benefits, particularly arising from guardians’ perceptions of the diets. Given the lack of large population-based studies, a cautious approach is recommended. If guardians wish to implement a vegan diet, it is recommended that commercial foods are used.

**Abstract:**

There has been an increase in vegetarianism and veganism in human populations. This trend also appears to be occurring in companion animals, with guardians preferring to feed their animals in accordance with their own dietary values and choices. However, there has been controversy amongst vets and online commentators about the safety of feeding vegan diets to carnivorous species, such as cats and dogs. In spite of this controversy, to date there has been no systematic evaluation of the evidence on this topic. A systematic search of Medline, Scopus, and Web of Science was performed, identifying 16 studies on the impact of vegan diets on cat and dog health. Studies were appraised for quality using established critical appraisal tools or reporting guidelines. There was considerable heterogeneity in the outcomes measured, and study designs employed, with few studies evaluating key outcomes of interest. Grading of Recommendations, Assessment, Development and Evaluation (GRADE) was utilized for assessment of certainty in the evidence, with the evidence for most outcomes being assessed as low or very low. Whilst the quality and amount of evidence needs to be considered in formulating recommendations, there was no overwhelming evidence of adverse effects arising from use of these diets and there was some evidence of benefits. It is, however, recommended that future high-quality studies, with standardized outcome measures and large sample sizes, be conducted. At the current time, if guardians wish to feed their companion animals vegan diets, a cautious approach should be taken using commercially produced diets which have been formulated considering the nutritional needs of the target species.

## 1. Introduction

In recent decades, the bond between humans and pets has led to an increased concern for pet health and welfare [1,2,3,4]. At the same time, veganism and vegetarianism has grown as an ethical and sustainable food choice. For example, according to a Vegan Society survey, the number of vegans in Britain increased fourfold between 2014 and 2019 [5].

This dietary regimen has been adopted not only by humans but also by their companion animals. However, the domestic cat (Felis catus) and domestic dog (Canis familiaris) belong to the order Carnivora, where cats are strict obligate carnivores [6] and dogs can be classified as facultative carnivores, both with anatomical and physiological characteristics that make them flesh-eating animals [7,8]. For example, their dentition is well-suited to crushing, cutting, grinding, and slicing meat [9], while their incisors, canines, premolars, molars, and carnassials are designed to hold their prey or pieces of meat [7]. They are also characterized by shorter, less complex digestive tracts with a lower capacity [10], since, unlike herbivores, their digestive processes do not require them to ferment food [11]. While dogs can eat plant-based foods, their anatomy preserves carnivorous traits of tearing muscle, crunching bones, and digesting meat more readily than plants [9]. In cats, all essential amino acids–especially taurine and arginine–need to be provided in the diet, and this is generally acquired through meat and animal-based products such as bone or viscera [12].

The dominant proteins in plant-based pet foods have historically been soy, corn protein, and wheat protein (gluten). Recently, additional plant proteins have become available for use, including pea protein, potato protein, and rice protein. Based on trends in human nutrition, this may expand further to include others such as hemp, oat, and bean proteins [13]. In contrast to animal tissues, plant cells are rich in carbohydrates (e.g., cellulose) that carnivores have difficulty digesting [11]. Proteins from cereal grains or soy contain lower amounts of essential amino acids. These include sulfur amino acids and the omega-3 fatty acids eicosapentaenoic acid and docosohexaenoic acid. Typically, they do not contain all essential vitamins, e.g., retinol (vitamin A) and cobalamin (vitamin B12) [12]. Additionally, plants may contain toxic compounds that only the gastrointestinal tract of herbivores can detoxify [11].

To date, there is still controversy over whether vegan and vegetarian diets can be recommended for dogs and cats [14]. However, most studies focus on the analytical composition rather than a clinical approach. According to the U.S. National Research Council’s (NRC) recommendations on nutrient requirements for dogs and cats [15,16], potentially problematic elements in vegan/vegetarian diets for dogs and cats could be: (1) insufficient protein; (2) unbalanced fats; and (3) nutrient insufficiencies [17]. For example, it has been shown that exercising dogs that consume unbalanced plant protein diets can develop anemia and a marked decrease in red blood cell hemoglobin levels but will return to health if the diet with vegetable protein is balanced properly [18]. Contrarily, meat-free diets for dogs and cats have been shown to be beneficial in treating certain diseases; for example, some urinary tract ailments [19] and cases of allergies caused by food proteins [20].

Given the controversy around the health-related aspects of feeding vegetarian diets to dogs and cats, this systematic review summarizes the available studies and findings regarding cat and dog health when fed vegan options. To date, there has been no systematic review of this topic using established methods of evidence synthesis. Knight and Leitsberger [15] performed a comprehensive narrative review on this topic using some elements found in systematic reviews. This previous review also evaluated studies on nutritional adequacy and should be referred to for this information. The current review updates this review and builds on the animal health-related aspects of studies.

## 2. Materials and Methods

### 2.1. Eligibility Criteria

Inclusive criteria were as follows: (P—population) studies that included dogs (*Canis familiaris*) or cats (*Felis catus*) in a domestic setting (construed broadly to include working dogs, as well as companion and laboratory animals). Studies evaluating animals of any breed, age or sex were eligible for inclusion; (I—intervention) studies where the animals had been fed a vegetarian or vegan diet (the latter containing no animal products—see definition below); (C—comparator) studies were included that compared the animals being fed meat-free diets, with those animals fed any kind of standard meat-based diets, or those studies without any parallel control group for comparison, e.g., within subjects comparison (pre-post diet change) or case-series design; (O—outcomes) outcomes had to be an animal-based measure that was related to animal health, physiology, or welfare. This could be a proxy report by the animal carer, in addition to those measures that could be measured directly upon animal examination; (S—study design) experimental and quasi-experimental study designs including randomized controlled trials, non-randomized controlled trials, and before-and-after studies were eligible for inclusion. Observational studies and case studies were also eligible for inclusion with critique related to study design being provided below in the results and discussion. This study is reported in accordance with PRISMA guidance [21].

### 2.2. Search Strategy

The search strategy aimed to locate published studies written in English and Spanish (the languages spoken by the authors). An initial limited search of Medline via Pubmed was undertaken to identify articles on the topic and any MeSH terms. The text words contained in the titles, abstracts, and index terms were used to develop a full search strategy for Medline. The search strategy was then adapted for the Scopus and Web of Science (including CAB Abstracts) databases. Key concepts were: “cat”, “dog”, “meat-free diet”, “vegan” “vegetarian”, “plant-based” and “health”. Grey literature, such as theses, sourced through these databases were eligible for inclusion. Reference lists of all studies selected for critical appraisal were screened for additional studies. The three databases were searched in October 2022 using the advanced search strategy derived (see Appendix A). There were no date restrictions on study selection, provided the studies were accessible through the three listed databases.

### 2.3. Study Selection

Following the search, all identified citations were collated and uploaded into Covidence (Veritas Health Innovation, Melbourne, Australia), and duplicates were removed. Titles and abstracts were screened by two reviewers (A.W. and A.O.) for assessment against the inclusion criteria for the review. Full text screening was similarly performed by two independent reviewers.

Any failures of consensus that arose between the reviewers at each stage of the study selection process were resolved though discussion, with a third reviewer (I.S.) being consulted if an extra opinion was required.

### 2.4. Assessment of Methodological Quality

Eligible studies were critically appraised for methodological quality using standard reporting guidelines/critical appraisal tools: a modified version of the SYRCLE risk of bias [22] assessment for randomized control studies, the STROBE-Vet statement for observational study designs (version adapted for veterinary studies) [23], and the JBI critical appraisal checklists for case reports (not veterinary specific) [24]. Survey studies were not critically appraised, since there are no published guidelines for doing this in studies of a veterinary nature which typically use animal guardians as respondents. This assessment was performed by two independent reviewers (A.W. and A.O.) using the quality assessment template created for these criteria in Covidence. Any disagreements that arose were resolved through discussion. All articles, regardless of the results of their methodological quality, underwent data extraction and synthesis. Consideration of the methodological quality of individual studies is discussed in the narrative synthesis.

### 2.5. Data Extraction

Data were extracted from included studies by three independent reviewers (A.W., A.O., and I.S.) with each study being extracted by two reviewers using a modified version of the Covidence 2.0 extraction template. Extracted data included specific details about the populations sampled, the study design, diets considered, and the outcomes relevant to health measured with the time-course of assessment relative to diet introduction. Any disagreements that arose were resolved through discussion.

### 2.6. Data Synthesis

Data were synthesized based on two subgroups: (1) species (dogs and cat studies) (2) direct assessment of measure of health versus guardian report, i.e., proxy measure. Within these subgroups, studies were categorized based on similarity in the nature of the outcomes measured. Due to the heterogenous nature of the studies, as well the limited number, it was decided by the review team that a meta-analysis was not appropriate for any data included in this review. There were generally few studies within the relevant subgroups and there was often a lack of a control group to compute the effect size. Clinical heterogeneity existed between studies in terms of the outcomes measured and the time since the introduction of the diet. As a result, the data are presented narratively in the form of tables and text.

### 2.7. Terminology

Within the veterinary literature, various terms have been used for plant-based diets and these are sometimes used without definition. This includes the terms vegan, vegetarian, meat-free and plant-based. Vegan diets are generally thought to refer to a complete absence of animal-based products in the diet (e.g., no egg or milk products), whilst vegetarian diets are normally considered to mean an absence of meat in the diet. For consistency, we have preferred to use the terminology “vegan diet” in this review to mean a complete absence of animal-based products but have reported the terminology used by authors of included studies in presenting our results. It is possible that these authors have not applied the terminology in the same way we have.

### 2.8. Assessing Certainty in the Findings

The approach of the Grading of Recommendations, Assessment, Development and Evaluation (GRADE) for grading the certainty of evidence was followed and a Summary of Findings (SoF) for each species was created using the GRADEPro GDT software (McMaster University, Hamilton, ON, Canada) [25].

## 3. Results

### 3.1. Description of Studies

Results of the search and screening process are presented in the PRISMA flowchart in Figure 1. Following the removal of duplicates and screening, 16 studies remained for inclusion. No additional studies were sourced outside the database search. The date of publication of these articles ranged from 1992 to 2022, with more publications investigating this topic in recent years (Figure 2). The majority of publications focused on dogs (11 out of 16), with six publications studying cats (one study [26] included dogs and cats). Study designs were varied with included studies consisting of a case report (n = 1), randomized control trials/experimental studies (n = 9), observational studies (n = 3), and survey studies of guardians (n = 9). Of the included studies, 13 directly measured health outcomes in the animals, whilst 8 gathered guardian reports on health outcomes or perceptions of health (two studies [27,28] performed both types of assessment). Refer to Table 1 and Table 2 for the characteristics of the included studies.

### 3.2. Feline Studies

Out of the six studies that considered cats, only four examined health outcomes directly via clinical examination or laboratory analyses of tissue samples (Table 1). Four studies used guardian proxy reports of health. For ease of reporting, the outcomes have been grouped based on their broad category. Meta-analysis was considered if more than one study presented the same outcome data. However, meta-analyses of these data were not possible due to (1) differences or lack of a comparison group, e.g., a meat-based diet comparator or (2) no presentation of a measure of central tendency or dispersion to input into the model.

#### 3.2.1. Hematology/Biochemistry

Only three studies [27,29,30] have carried out hematological and/or biochemical analysis of blood in cats that were fed vegetarian diets, and it is worth noting that sample sizes were low. Cats on a high-protein vegetarian diet exhibited hypokalemia which accompanied recurrent polymyopathy [29]. There was also increased creatinine kinase activity, likely reflecting the muscle damage caused by the myopathy, and reduced urinary potassium concentrations. Potassium supplementation prevented development of this myopathy, strongly suggesting a link between the potassium and myopathy. Although, interestingly, spontaneous recoveries of myopathy in the non-supplemented groups were not consistently associated with increases in plasma potassium. Whilst urea levels were slightly above the laboratory reference range, there was no change in levels in either supplemented or non-supplemented animals across the time-course of the 6-week dietary treatment. Biochemical findings in other studies have generally been unremarkable [27] with normal serum iron, total protein and albumin [30].

A macrocytic, non-regenerative anemia was observed in both felines that were presented in the case study of Fantinati et al., 2021 [30]. Otherwise, hematology was generally unremarkable.

#### 3.2.2. Amino Acids/Specific Biomarkers

Leon et al., 1992 [29] showed that plasma taurine concentrations decreased by approximately 87% after only 2 weeks on a vegetarian diet (from 122 μmol/L to 16μmol/L). By the end of the 6-week study, there was no detectable taurine in plasma. Taurine concentrations were not different between the potassium-supplemented and non-supplemented groups, with both groups showing this substantial drop in taurine. Glutamate also increased in both groups but appeared to stabilize after 2 weeks of diet feeding [29]. Conversely, in the two cats described in the case report of Fantinati et al., 2021 [30], despite a 5-month duration of feeding of a vegan diet, taurine and cobalamin were within reference ranges. These findings were largely corroborated by the study of Wakefield et al., 2006 [28], where 17 cats on long-term vegetarian diets were blood sampled. Out of the 17 cats, blood taurine was within reference range for all except three, for whom taurine fell below the reference range, although not critically so. Similarly, cobalamin has also been shown to mostly fall within reference range [27,28]. It is, however, worth noting that in Semp 2014 [27], most of these animals were on commercially available supplements.

Serum folate was shown to decrease substantially in the two cats described in the Fantinati et al., 2021 case study [30], the correction of which improved clinical signs. Folate was decreased in approximately 50% of cats (n = 8) and was significantly so in some cats (approximately half of the reference range values), in another study [27]. Folate was not measured in the Wakefield et al., 2006 study [28].

#### 3.2.3. Clinical Findings

In cats fed vegetarian diets that were supplemented with potassium, a myopathy was seen within 2 weeks of the dietary change [29]. This was characterized by ventroflexion of the head and the neck. The cats also showed lateral head resting, a stiff gait, muscular weakness, unsteadiness, and the occasional tremor of the head and pinnae. Erythrocyte transketolase activity was assessed to determine whether thiamine deficiency was contributing to the clinical myopathy, independent of potassium status. Differences in this enzyme across the time-course of the study were non-significant, suggesting thiamine deficiency was not a causative factor in the development of the clinical signs. Thiamine was also found to be within the reference range in Fantinati et al., 2021 [30]. No abnormalities were detected on auscultation or ophthalmoscopic examination [29]. Weight loss and poor coat condition have also been observed in cats fed vegetarian diets [29,30]. However, most cats in another study had a normal coat condition and no obviously diet-related clinical abnormalities picked up by clinical examination [27]. Clinical signs of lethargy with altered mentation, dysorexia, and muscle wasting, along with gut signs of bloating and increased borborygmi have also been observed [30]. Yet, the defecation of cats on vegan diets has been shown to be unremarkable [27].

#### 3.2.4. Guardian-Reported Health Effects

Guardians generally believed that the transition to a meat-free diet had been positive. These studies are valuable, as large sample sizes of respondents (animals) are generally employed. Some guardians did notice an increase in stool volume but noted no issues with consistency [27]. When considering other aspects, coat condition was shinier [27], there was an improved scent of their animals (particularly relating to breath odor) [27], there was a tendency to be at the ideal body condition score rather than being obese [28,31]. Lifespan was also not considered to be influenced by diet [31]. Palatability, which can influence an animal’s mental health was reported by guardians to not be an issue with vegan diets [26].

Dodd et al. (2021) [31] collected dietary information for 1026 cats, of whom 187 were fed vegan diets. The latter were more frequently reported by guardians to be in very good health. They had more ideal body condition scores and were less likely to suffer from gastrointestinal and hepatic disorders than cats that were fed meat. No health disorders were found to be more likely in cats that were fed vegan diets. The reported differences were statistically significant.

**Table 1 vetsci-10-00052-t001:** Characteristics of studies evaluating vegan diets in cats.

Author	Signalment	Study Design	Sample Size	Diet Type	Duration of Feeding Diet	Comparator	Outcomes Assessed
**Animal—Based Outcomes**							
Fantinati et al., 2021 [30]	2-year-old male neutered Main Coon and a 1-year-old female spayed Domestic Shorthair cat,	Case Report	2	Commercial plant-based pet food	5 mo	N/A	Clinical examination Body condition Blood hematology/biochemistry Folate (vitamin B9) Taurine Thiamine iron and serum ammonia Food intake
Leon et al., 1992 [29]	M, F	Randomized controlled trial	5	High protein vegetarian diet	6 weeks	Compared vegetarian diet versus vegetarian diet with potassium supplementation	Clinical examination Body weight Blood biochemistry Erythrocyte transketolase activity Taurine Glutamine Potassium levels
Semp 2014 [27]	Not stated	Cross-sectional	15	Commercial or homemade vegan diet	Average 3.9 years, range 6 mo- 6.5 years	N/A	Clinical examination Blood haematology/biochemistry Folate Cobalamin (Vitamin B12)
Wakefield et al., 2006 [28]	M,F Average age = 6.6 ±4.4	Cross-sectional	17	Home-prepared vegetarian diet with majority using a commercially available supplement	Average 4.4 ±3.4 years	Standard lab reference ranges	Taurine Cobalamin (Vitamin B12)
**Guardian—Reported Outcomes**							
Dodd et al., 2021 [31]	M,F Mean age 7.5 ± 4.85 Mixed breeds: DSH = 660 DMH = 60 DLH = 95 Asian = 50 American = 35 Other/unknown = 154	Survey study- cross sectional of cat guardians	1325 guardians 1026 cats	Most cats had a meat-based diet n = 65%, 18% Plant Based diet, 6.7% combination. 10% indeterminable. Supplements fed to 19% being more often used when cats had PB diet (40%)	Average 3.6 years for both diets	Responses of owners feeding meat-based diet compared with plant-based diet	Body Condition Fecal Score Health Conditions Guardian’s perception of health Lifespan (indicated for previously owned cats)
Knight and Satchell 2021 [26]	Various, unspecified	Cross sectional survey	Guardian reports on 1135 cats	Conventional meat diet, raw meat or vegan pet food			Guardian reported palatability/feeding behavior
Semp 2014 [27]	Various, unspecified	Cross sectional survey	59 cat guardians	Commercial vegan food or self-prepared vegan diet	Average of 3.9 years	N/A	Guardian reported health
Wakefield et al., 2006 [28]	M,F Mean age 7 years	Cross sectional telephone survey	Vegetarian diet = 34 Meat-based diet = 52	Home-made vegetarian diet and conventional meat-based diet	At least 1 year	N/A	Guardian reported body condition and perceived health status

MF: male, female.

### 3.3. Canine Studies

Twelve studies examined outcomes in dogs that were fed vegan diets, with nine studies measuring outcomes directly in the animals (Table 2). Four studies used guardian proxy reports of health. Findings from the studies evaluating outcomes in dogs have been grouped based on animal or guardian-based measures and are reported in Table 3 and Table 4, showing the direction of effect for the indicators assessed. As for the feline studies, meta-analyses were considered if more than one study presented the same outcome data. Unfortunately, this was not possible for any of the outcome measures due to the lack of a comparator group within studies that may have contributed data.

**Table 2 vetsci-10-00052-t002:** Characteristics of studies evaluating vegan diets in dogs.

Author	Signalment	Study Design	Sample Size	Diet Type	Duration of Feeding Diet	Comparator	Outcomes Assessed
**Animal—Based Outcomes**							
Brown et al., 2009 [32]	Pure-bred Siberian huskies Racing sled dogs Age not stated	Non-randomized experimental trial	12	Meat-free diet	16 weeks	Compared a commercial meat-based diet with an experimental meat-free	Food intake Body weight Complete blood count
Cavanaugh et al., 2021 [33]	PB diet Median age = 2.9 years Median BW = 19.5 kg Mixed breed: 27 Siberian husky: 2 Pitbull: 2 Border collie: 1 Vizsla: 2 Pug: 1 Spayed females: 16 Castrated males: 1 Intact female and male: 1 Control diet: Median age = 3.8 years Median BW = 20.8 kg All mixed breed and spayed females and males	Non-randomized experimental study	PB = 34 Control = 4	Commercial plant-based diet Control diet (traditional dry commercial diet)	12 weeks 4 weeks wash in period	Compared to a traditional dry commercial diet	Clinical sign Biochemical profile Plasma amino acid panel Taurine Urinalysis Echocardiography Guardian-based acceptance
Cavanaugh et al., 2022 [34]	Adult mixed-breed dogs G1 and G2 average age = 3 and 2.8 years, respectively G1 and G2 average weight = 14.3 and 14.8 kg, respectively	Randomized controlled trial	16	Commercial extruded plant-based diet	4 weeks	Compared to commercial extruded traditional diet	Physical examination Blood analysis for TMOA, choline, betaine, and creatine
El-Wahab et al., 2021 [35]	Healthy female Beagle Average BW = 11.0 ± 1.31 kg Average age = 3 years	Experimental crossover design	8	Vegetarian basic diet Vegetarian basic diet with hydrolyzed feather meal and either corn meal, fermented rye or rye supplemented	10 days	N/A	Food and water intake Body weight Fecal score Digestibility
Ingenpaß et al., 2021 [36]	Unneutered female Beagle dogs Average BW = 9.64 ± 0.68 kg Average BCS = 4.98	Experimental crossover design	6 (n = 3 per diet)	Vegetarian diet containing wheat gluten (8.81%), rice protein (8.81%), sunflower oil (6.84%) and vitamin D3 (0.045 g/kg)	24 days	Compared to meat-based diet containing poultry meal (19.5%) and poultry fat (5.23%)	Fecal quality Apparent digestibility Crude protein Crude fat Nitrogen estimation
Kiemer 2020 [37]	Average age = 2.15 years Whippet, Golden retriever, mixed, Australian mini shepherd, cocker-mixed, German shepherd cross, Husky, Boston terrier, Daschund, Cavalier king Charles spaniel, French bulldog, Corgi	Randomized controlled trial	PB diet = 20 Control = 20 Feeding trial = 8 (vegan = 4, control = 4)	100% vegan diet for at least 3 months before the trial	6 weeks	Compared to meat-based diet	Blood chemistry Physical examination Guardian survey
Semp 2014 [27]	Not stated	Cross-sectional	20	Commercial plant-based pet food 27% supplemented diet 65% no supplemented diet	Average length of diet consumption: 2.83 years	N/A	Clinical examination Body condition score Blood hematology/biochemistry Pancreatic parameters Magnesium Calcium Total protein Folic acid Vitamin B12 Carnitine
Rankovic et al., 2020 [38]	Siberian husky Male neutered (n = 4) Female spayed (n = 5) Female intact = 2 Average age = 5.63 ± 0.57 Average BCS = 4.80 ± 0.66 Average BW = 24.94 ± 0.99 kg	Experimental crossover design	Study 1 = 6 Study 2 = 11	Three starch rich foods: white bread, cooked white long grain rice, and cooked Eston green lentils	150 min	Four commercial foods	Glycemic index Glycemic response Insulinemic response
Richards et al., 2021 [39]	Adult, client-owned Siberian huskies Neutered males = 4 Spayed females = 5 Intact females = 1 Average age = 5.63 ± 0.72 years Mean BW = 23.32 ± 1.15 kg	Non-randomized experimental	11	Vegetarian formula	Experimental day and 7 weeks wash out period	Four commercial foods	Gastric emptying
**Guardian—Reported Outcomes**							
Dodd et al., 2022 [40]	Average age = 6 years 679 males 688 females MB diet = 665 PB diet = 357 118 specific breeds 162 crossbreds	Qualitative research	1189	Plant-based diet	9 months	Responses were compared to those of guardians feeding meat-based diets and to a combination of both	Guardian perception of health and wellbeing BCS Lifespan (previously owned dogs)
Knight and Satchell 2021 [26] *	Females 82%	Survey	2639 dogs (4057 guardians of dogs and/or cats)	Vegan	Mixed survey	Conventional diet Raw meat diet Mixture	Guardian -reported palatability
Knight et al., 2022 [41] *	Fed a conventional meat diet (54%) Raw meat diet (33%) Vegan diet (13%) Vegan diet animals with health disorders (36%)	Cross sectional survey	2639 dogs	Vegan diet	At least 1 year	Comparison between the three types of diets (meat, raw meat, and vegan)	Veterinary visits Medication use Guardian based health Seven general health disorders 22 specific health disorders
Semp 2014 [27]	M,F Not stated	Cross-sectional survey	38 guardians 174 dogs	Commercial plant-based pet food 27% supplemented diet 65% no supplemented diet	Average length of diet consumption: 2.83 years	N/A	Guardian-reported health Stool consistency Coat condition

BCS: body condition score; BW: body weight; MF: male, female; MB: meat-based diet; PB: plant-based diet. * These studies relate to the same survey dataset with different outcomes presented in the two studies.

#### 3.3.1. Hematology/Biochemistry

Five studies performed hematological and biochemical profiles in dogs to assess essential amino acids, vitamin, mineral, and folic acid levels when compared to a meat-based diet, or after changing to a vegan alternative. Brown et al., 2009 reported an increase in erythrocyte count, hemoglobin, and packed cell volume, although these stayed within reference ranges without signs of anemia or health disorders [32]. Higher hemoglobin values were also reported in 28 dogs by Cavanaugh et al., 2021 [33]. In this same study, one dog had alanine aminotransferase, and two dogs had alkaline phosphatase levels above reference values. Serum total protein and albumin were all within the reference range in another study [27].

#### 3.3.2. Amino Acids/Specific Biomarkers

Cavanaugh et al., 2021 [33] assessed essential amino acids and taurine levels after 12 weeks on a commercial extruded plant-based diet with pea protein as the main protein source, with the finding of lower (but within or above reference range) leucine, methylhistidine-3, and serine. For this reason, the authors concluded that a plant-based diet can provide the necessary elements to dogs. Similarly, although choline and betaine, two precursors of trimethylamine-N-oxide (TMAO), decreased in sixteen mixed-breed dogs, it did not influence the health of animals [33].

In Kiemer’s 2020 study, blood sampling showed that only two animals that were fed vegan diets out of 20 had any deficiencies [37]. This related to folic acid but was considered to be a response associated with giardiasis. Contrarily, 11 animals fed meat-based food had decreases in some parameters. Parameters where there were significant differences between meat and vegan groups were iron, vitamin B12 and folic acid, although it is worth noting that, whilst statistically different, these elements did not move outside of reference ranges. Values within normal reference ranges were also described for serum folic acid (71% of 17 dogs), vitamin B12 concentration (an average of 340.09 pg/mL), iron (within reference values in 90% of dogs), and L-carnitine in 57% of dogs on vegan diets in another study [27].

#### 3.3.3. Clinical Findings

Dogs fed vegan diets were shown to be in ideal body condition, with normal behavior and skin/coat condition [37]. Similarly, no muscle loss or poor haircoat was detected in a further 28 dogs in the study by Cavanaugh et al., 2021 [33], with no echocardiographic differences between animals receiving vegan and conventional diets [33].

Experimental trials with beagle dogs fed vegetarian alternatives showed normal fecal scores and high acceptance of the diet [35].

Whilst vegetarian diets may increase the nitrogen content excreted in feces, this is not significant when compared to the traditional diet [36]. The glycemic index of the vegan diet was also no different to that of other starch diets [38], and with this type of diet, gastric-emptying was higher than in traditional diets (0.0123 kSB2,/min vs. 0.0010 kSB2,/min) [39].

A change that has been reported in studies where dogs received a vegan diet is the increase in urinary pH. In a study with 19 dogs, one animal had urinary pH above the normal range (pH 5–7); however, no clinical signs were associated with the alkalinity [27]. Cavanaugh et al., 2021 similarly reported that after 12 weeks of a commercial plant-based diet, urine pH was higher (*p* = 0.022), considered to arise from the lack of protein of animal origin [33].

#### 3.3.4. Guardian Perception

In general, guardians perceive vegan diets as a beneficial alternative for dogs with a positive impact on their health [37]. This is reflected in 57% of guardians reporting an ideal body condition score, as well as longevity, fewer health disorders, and very good health in their dogs on vegan diets [40]. Based on guardian-reported health status, Knight et al., 2022 [41] suggest that, when compared to meat and raw meat diets, vegan options are the healthiest and least hazardous choices for companion dogs, making dogs less likely to need veterinary visits more than once per year, and leading to less medication use than other animals [41]. In another study from the same group [26], palatability was the main assessed parameter. Palatability can be associated with mental health, maintenance of weight, and therefore the promotion of welfare, rendering this outcome valid as a health finding. This study showed no difference in food intake, i.e., change in palatability with a vegan diet compared to meat-based diets [26]. This aspect is also an important consideration when owners are deciding whether to transition their animals onto new diets.

Dodd et al. (2022) [40] collected dietary information from 1189 dog guardians, including 357 implementing solely vegan diets, who fed these diets for 3 years on average. Vegan dogs were reportedly more likely to experience very good health, and less likely to suffer ocular, gastrointestinal and hepatic disorders. No health disorders were more likely, and longevity of previously owned dogs was reportedly 1.5 years greater, when fed purely vegan diets. These reported differences were statistically significant.

Knight et al. (2022) [41] studied 2536 dogs fed vegan or meat-based diets for at least one year. Included were guardian opinions of health—which are not always reliable—as well as a range of more objective data, such as prevalence of medication usage. Seven general health indicators were examined, along with the prevalence of 22 of the most common canine health disorders. The researchers concluded that the healthiest and least hazardous diets for dogs were nutritionally sound vegan diets. Some differences were statistically significant.

### 3.4. Risk of Bias of Included Studies

A summary of the methodological quality assessment for the included articles is provided below, split by study type for those studies that used quantitative methods. Survey studies were not appraised, given the lack of established critical appraisal tool for this type of research within veterinary medicine. It is worth noting that the tools used for the observational and case study designs are strictly reporting guidelines rather than checks for risk of bias and are presented as a general indication of quality but do not necessarily predict internal validity of the study.

#### 3.4.1. Randomized Control Trials

Nine studies were assessed as randomized control trials (Figure 3 and Figure 4). Crossover studies and studies where the authors categorized them as non-randomized studies were also assessed as RCTs, since it was deemed that randomization could occur, or at least there were methods to reduce the risk of selection bias. An unclear rating was assigned when criteria were not applicable for that study or when there was a lack of reporting; for example, on randomization. A high risk was assigned where there was reporting on a criterion and the methods used were considered to introduce a risk of bias; for example, if randomization was not performed by a method that would achieve true randomization. Overall, the studies were generally evaluated as being at an unclear risk of bias. There was a consistent lack of reporting on any methods to achieve randomization, allocation concealment, or blinding at the performance or outcome levels. There was generally a low risk of bias for attrition and selective reporting. The latter is, however, hard to assess when protocols are not pre-registered. Hence a conversative approach was taken, considering that so long as the outcomes discussed in the methods had been reported, the risk of bias was low. Where an unclear result arose for the category of ‘other risk of bias’, this generally related to funding by a pet food company with no discussion of how any potential conflicts of interest were minimized.

#### 3.4.2. Observational Studies

Two studies were assessed as observational studies using reporting guidelines (Figure 5 and Figure 6). Reporting was generally good. The main issues of lack of reporting (illustrated in red) arose around determination and reporting of how a sample size was arrived at, and discussions around transparency, including the role of funders, and whether ethics committee approvals were obtained.

#### 3.4.3. Case Studies

Only one case study was included in the review [30]. This was appraised using case study reporting guidelines, and this study was rated as having a low risk of bias, i.e., reported all the criteria assessed (Figure 7 and Figure 8).

### 3.5. GRADE Certainty Assessment and Results

The evidence presented across the main parameters of interest, and those parameters where there were the most investigations, was assessed using the GRADE approach. Results are presented by species in Figure 9 and Figure 10. The certainty in the evidence was graded as low or very low for all outcome parameters, with the exception of bodyweight/condition in dogs, where certainty in the evidence was graded as moderate. Evidence was often downgraded based on the risk of bias as described previously, with the predominance of observational study designs. There was occasionally inconsistency in the direction of effect of outcomes across studies. Imprecision was generally regarded as not serious. The studies were also considered to be direct, having been conducted in the species of interest, in a clinical scenario. Publication bias has not been formally assessed and was rated as undetected. Publication bias may not be a factor for this topic since the subject is generally of interest, and there is little existing or key literature that may conflict with newer understandings.

## 4. Discussion

The finding of this study suggests, on the face of it, that there is very little evidence of major adverse effects resulting from the feeding of vegan diets in dogs or cats. The majority of the animal-based parameters were within normal reference ranges and when there were deviations from normal reference ranges, there were rarely clinical signs reported alongside the finding. In addition, whilst the broad literature in this area commonly makes reference to concerns around nutrient deficiencies, such as that of taurine, folate, and cobalamin, there were a limited number of studies that measured these outcomes (generally, only two studies for key outcomes), with limited evidence of these deficiencies arising (with some of the alterations likely being attributable to confounding; for example, as a result of secondary disease, e.g., giardiasis in a dog). These conclusions should, however, be interpreted cautiously, given the breadth and quality of the evidence presented as described below.

### 4.1. Evidence Considerations

To date, only sixteen studies have looked at actual health-related outcomes in dogs and cats fed vegan diets, as opposed to performing nutrient evaluations of diets. However, the majority of these studies utilized small sample sizes (ranging from 2–34 animals) for the direct investigation of outcomes. Whilst survey studies evaluating guardian-reported outcomes generally encompassed larger numbers of animals, these are subject to inherent biases due to participant selection, as well as the reliability of lay people making judgements around somewhat subjective concepts, such as health and body condition. Whilst 9 out of 13 of the studies that directly measured outcomes in the animals employed study designs which sit high within the evidence hierarchy, such as randomized controlled trials or experimental studies, the limited sample sizes and challenges inherent in crossover designs, such as choosing suitable washout periods, does limit certainty in the findings of these studies.

It is also worth noting in relation to the studies that measured animal health directly that, with the exception of two studies [27,28], the dietary intervention was often short being in the order of weeks to months generally, rather than years. On short periods such as these there may not have been time for deficiencies to develop or for clinical signs to become apparent.

The risk of bias assessment performed on the experimental trials suggests, at best, an unclear risk of bias across the studies. There were some particular aspects of poor performance (or reporting), especially around randomization and blinding. This has been reported previously in animal studies [42], where researchers have probably not taken on board some of these important facets of experimental design and reporting to the extent that human clinical researchers have [43,44]. This remains a major concern impeding reproducibility, and where internal validity of the study is impacted, also leads to wastage of animal and financial resources [42].

It was not possible to perform meta-analytic techniques on the data as described previously. There was also considerable heterogeneity in the outcomes measured. Taken together, this limits the ability to make conclusions and recommendations for practice since statistical assimilation of data does increase the value of the evidence synthesis. This evidence base points to the need for future research to employ larger sample sizes and to perform, as a priority, direct animal-based studies to generate firm conclusions around the suitability, or otherwise, of vegan diets in dogs and cats.

### 4.2. Guardian Perceptions

A change to vegan diets for both guardians and their animals is generally driven by increased guardian concerns about health, welfare of production animals, and environmental sustainability [17]. A small proportion of guardians may make a dietary change due to their animal not accepting traditional diets [27]. Whilst surveys of guardians who adopt this nutritional lifestyle for themselves may be subject to an inherent selection bias due to their strong ethical convictions, it is clear that guardians generally perceive these diets as beneficial. Guardians believed that it was easier to maintain their animals at an ideal body condition score, and there was a general perception of animals having fewer health disorders, and less need for veterinary visits. For example, guardians in Dodd et al.’s study (2021) [31] reported that 52% of cats did not have health disorders, and their analysis found that cats fed a vegan diet, compared to animals consuming a meat-based one, had less prevalence of dental (21 vs. 131, respectively), gastrointestinal and hepatic (3 vs. 90), and ocular diseases (4 vs. 39). A similar outcome was observed in dogs in another study by the same authors, where the decision to provide a vegan option to the dogs resulted in a reduced prevalence of several disorders such as cardiac, dermatopathies, and renal issues [40]. Another key benefit was less odor from their animals. The only negative effect reported, which seemed to be considered only a minor inconvenience by guardians, was an increase in stool volume. This increase was not accompanied by any change in stool consistency.

It is perhaps unusual, given the rise of veganism in society [5], that more largescale studies on the health consequences of these diets in companion animals have not been conducted, since it is assumed that there would be a similar increase in the use of these feeding regimes for pets. There is a dearth of primary data on the proportion of pet guardians that feed vegan diets. Dodd et al., 2019 [45] derived figures of 1.6% of dogs and 0.7% of cats in a global online survey, with guardians being predominantly vegan themselves. There does, however, appear to be an increasing commercial need with three times as many vegetarian pet foods being launched in the UK in 2014, compared with the previous three years [46]. There is also evidence of interest in converting pets to a vegan diet with 35% of surveyed guardians who did not feed their pets a vegan diet expressing interest should certain conditions be met [45]. Knight et al.’s (2022) [41] study surveyed 1370 respondents who used a conventional meat formulation as their dog’s normal diet, and 830 who used a raw meat formulation. These combined 2200 respondents were asked whether they would realistically choose alternative diets if such diets offered their desired attributes. The alternatives offered for consideration were vegetarian and vegan diets, as well as those based on laboratory grown meat, insects, fungi, and algae. Of 2181 who answered this question, 44% (955) confirmed they would realistically choose such alternative diets. Barriers to converting to vegan diets included a need for further information on nutritional adequacy, and a requirement for veterinary approval, and greater availability of commercial diets [45]. It is not clear where the veterinary profession stands on the feeding of vegan foods to dogs and cats, with veterinary professional associations appearing to be silent on the issue, deferring to individual veterinary advice (see e.g., [47]). The situation likely reflects the myriad of viewpoints held by individuals within society, but it is quite possible that the advice of veterinarians is a significant barrier to diet conversion. This advice may also be based on outdated education or lack of understanding of the evidence available.

### 4.3. Cats Versus Dogs

Due to the carnivorous physiology of dogs and cats, there are concerns that various mineral, vitamin, and amino acid deficiencies may arise if they are fed plant- based diets. Concerns have particularly been raised around sulfur amino acids, methionine, L-carnitine, cysteine, phosphorus, calcium, vitamins D and B12, omega-3 fatty acids, taurine, and arachidonic acid [48].

There has been particular concern around the ability to feed cats vegan diets, since they are regarded as obligate carnivores which require certain nutrients in their diet (based on their inability to manufacture these themselves). Figure 11 depicts some of these essential nutrients and the health issues that have been discussed to arise in their absence. This review provides limited evidence for adverse health impacts arising in cats fed vegan diets, although this needs to be considered in light of the small number of studies performed and often limited sample sizes. Major concerns were only noted around deficiencies in taurine and folate, and these were not seen across all cats sampled, suggesting local factors may have been at play, such as that created by dietary variations. It is, however, also worth noting that in a number of the feline studies, cats were supplemented, e.g., [31] and it may be that this supplementation avoided any adverse consequences. The issue of supplementation is important and we did not review the suitability of supplements specifically in this review. Perhaps a take-home message is that use of commercially prepared vegan pet foods appear to be safe for use in cats and dogs but further research is needed.

This field is attracting more attention by researchers, as well as the pet food industry, with new studies regularly being published. In fact, a recent study was sourced after the formal searches for this review. Davies (2022) [49] evaluated the health perception of 100 guardians providing extruded vegan food to dogs. The findings were overtly positive with the food being palatable and leading to no adverse changes in appetite or body weight. Health improvements were also reported by the guardians in the following areas: activity level (28%), fecal consistency (38.5%), antisocial smelling flatus (73.1%), coat glossiness (49.0%), itching (60.6%), skin redness (44.4%), and even in behavioral traits such as aggression (25.0%) and coprophagia (42.9%). It is hoped that, within the next few years, recommendations on this topic may be able to be made with a greater degree of certainty, given this research interest.

## 5. Conclusions

This review has found that there is no convincing evidence of major impacts of vegan diets on dog or cat health. There is, however, a limited number of studies investigating this question and those studies available often use small sample sizes or short feeding durations. There was also evidence of benefits for animals arising as a result of feeding them vegan diets. Much of these data were acquired from guardians via survey-type studies, but these can be subject to selection biases, as well as subjectivity around the outcomes. However, these beneficial findings were relatively consistent across several studies and should, therefore, not be disregarded.

There is an urgent need for large-scale population-based studies to further investigate this question, with a particular focus on assessing the dietary aspects cited to be of particular concern, e.g., taurine and folate. For guardians wishing to feed their pets vegan diets at the current time, based on the available evidence it is recommended that commercially produced vegan diets are used since these are less likely to lead to nutrient imbalances.

## Figures and Tables

**Figure 1 vetsci-10-00052-f001:**
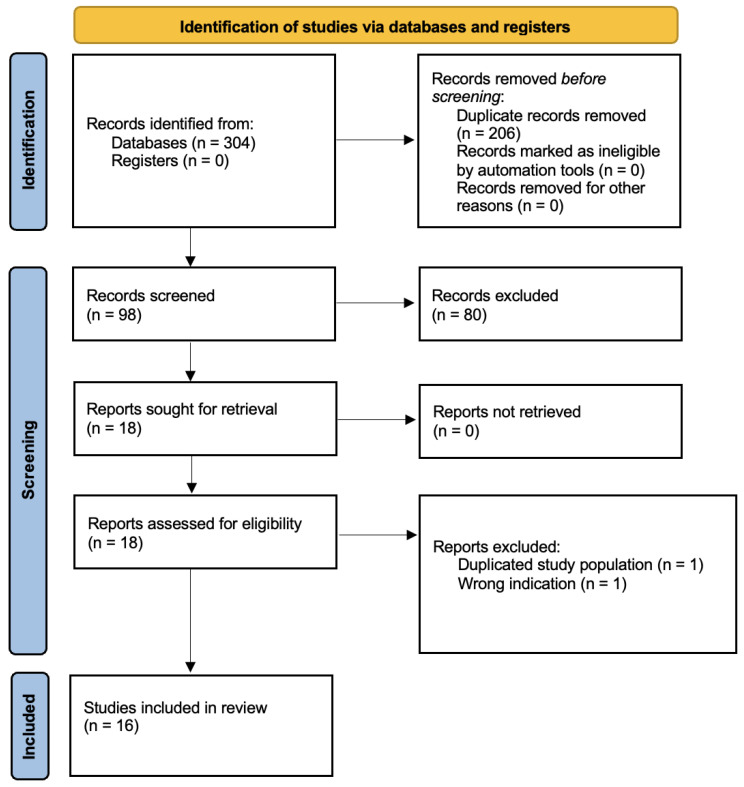
PRISMA [21] flow diagram for the systematic review detailing the databases searched, the number of abstracts screened, and the full texts retrieved.

**Figure 2 vetsci-10-00052-f002:**
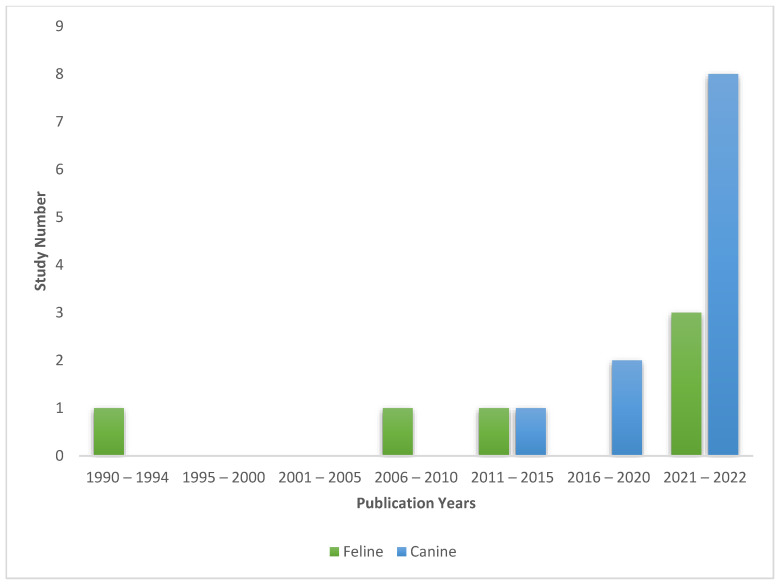
Bar chart illustrating number and years of publication for studies involving dogs and cats fed vegan diets.

**Figure 3 vetsci-10-00052-f003:**
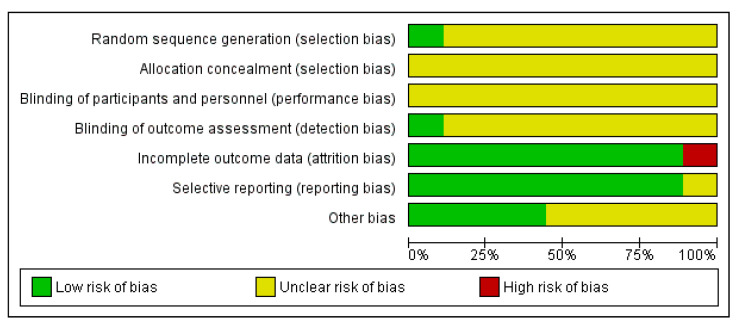
Risk of bias graph: review authors’ judgements about each risk of bias item presented as percentages across all included studies.

**Figure 4 vetsci-10-00052-f004:**
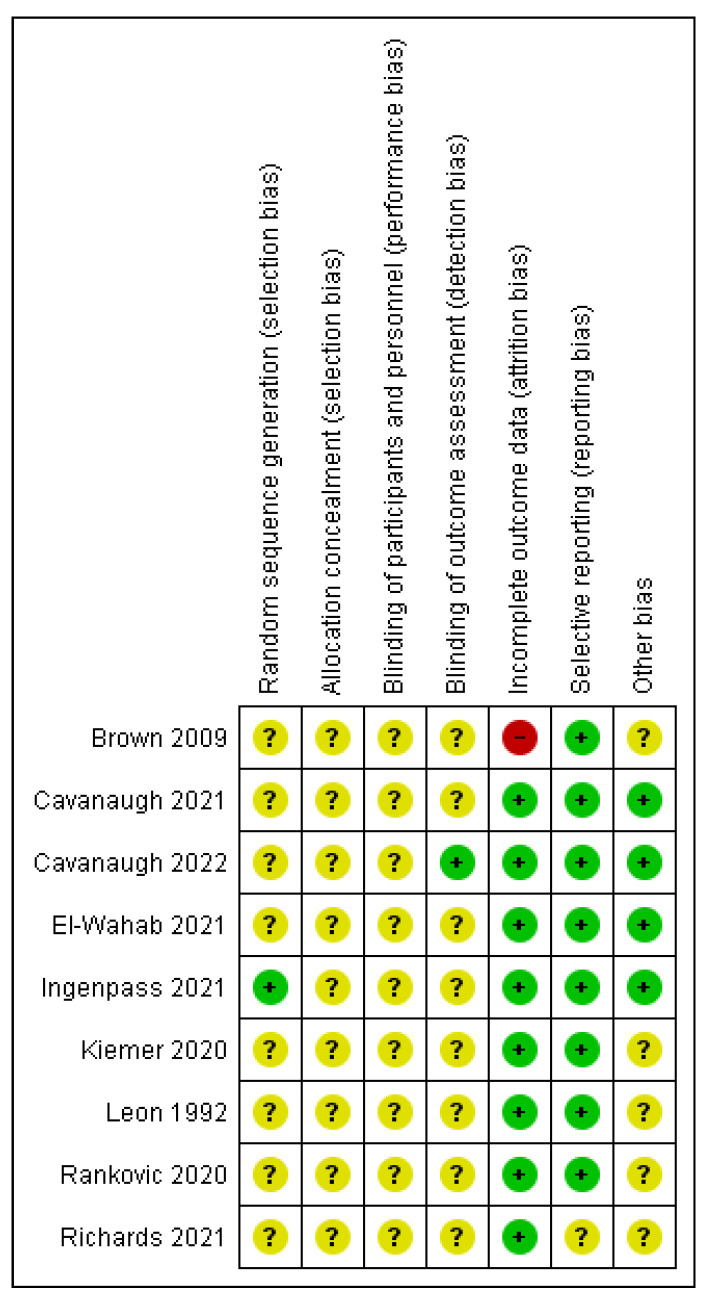
Risk of bias summary: review authors’ judgements about each risk of bias item for each included study [17,29,33,34,35,36,37,38,39].

**Figure 5 vetsci-10-00052-f005:**
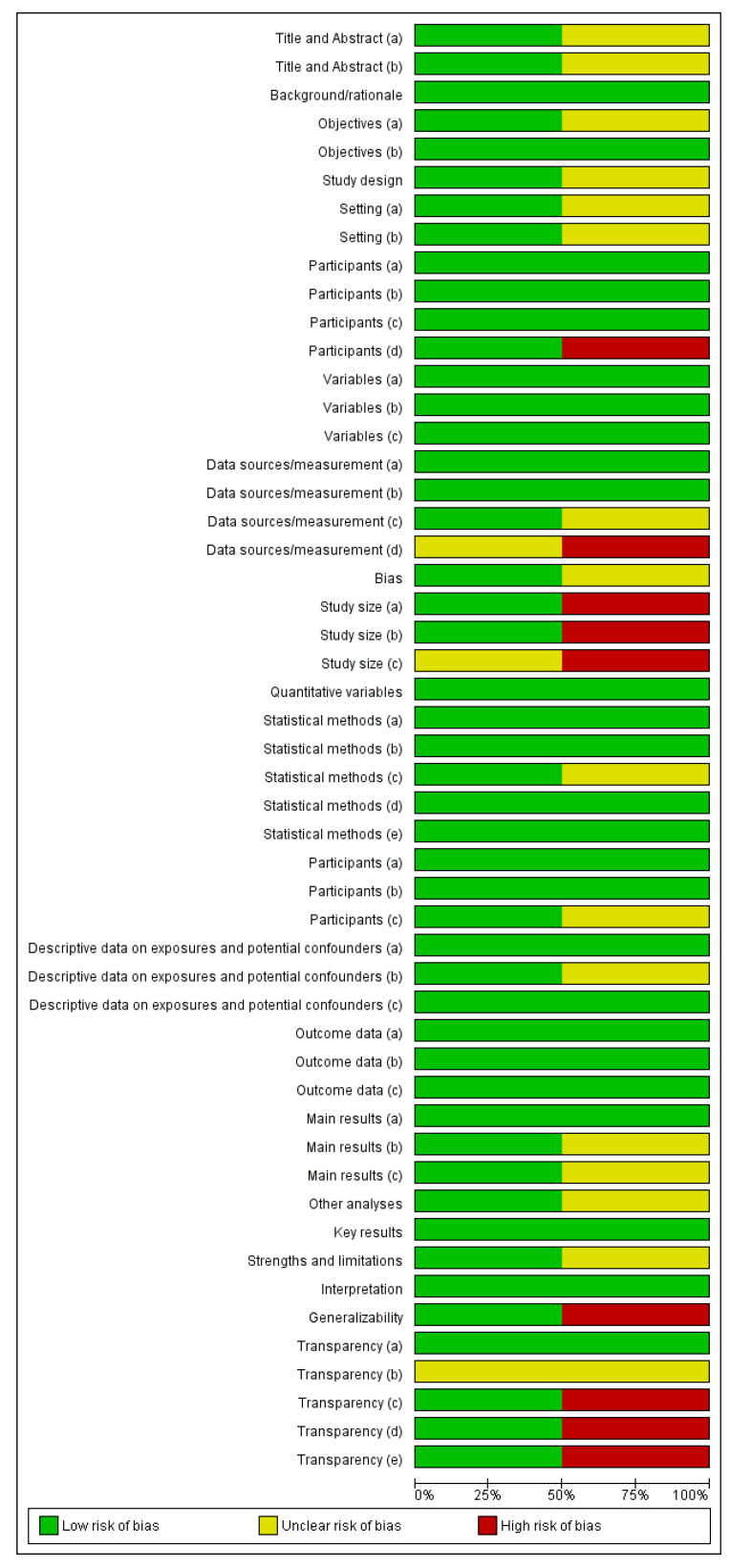
Reporting guidelines summary: review authors’ judgements about each criterion presented as percentage across all included studies.

**Figure 6 vetsci-10-00052-f006:**
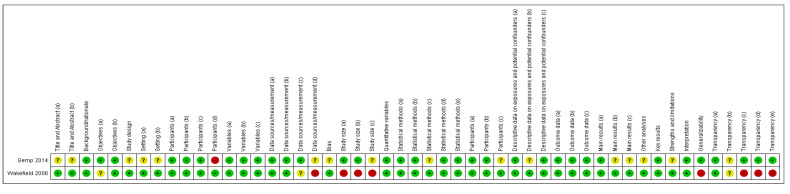
Reporting guidelines summary: review authors’ judgements about reporting on each criterion for each included study [27,28].

**Figure 7 vetsci-10-00052-f007:**
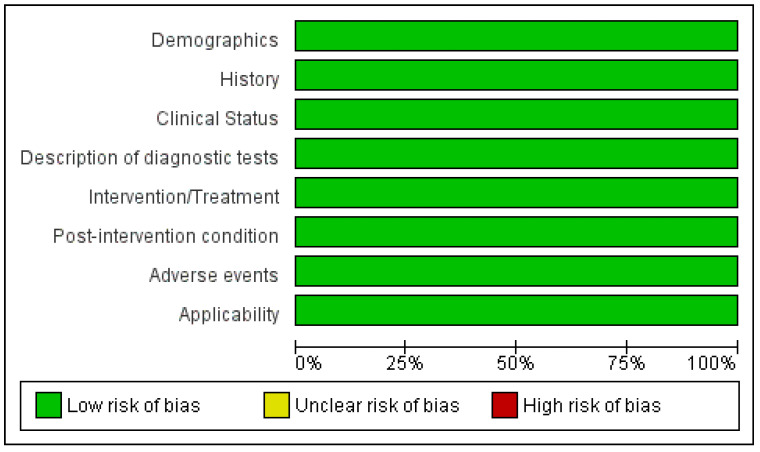
Reporting guidelines summary: review authors’ judgements about reporting on each criterion as percentage across all included studies.

**Figure 8 vetsci-10-00052-f008:**
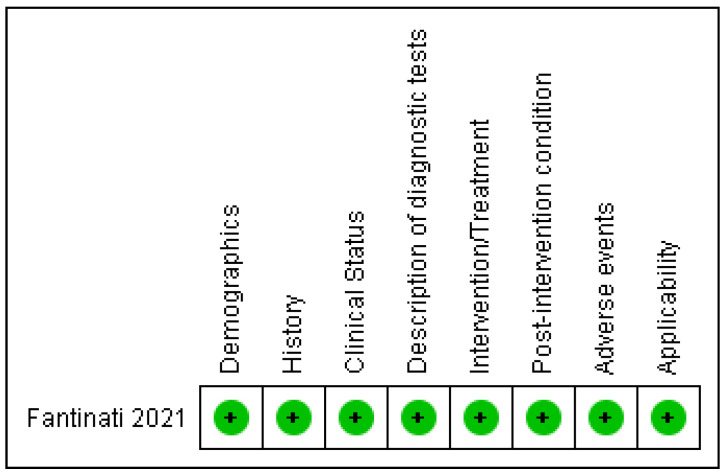
Reporting guidelines summary: review authors’ judgements about reporting on each criterion for each included study.

**Figure 9 vetsci-10-00052-f009:**
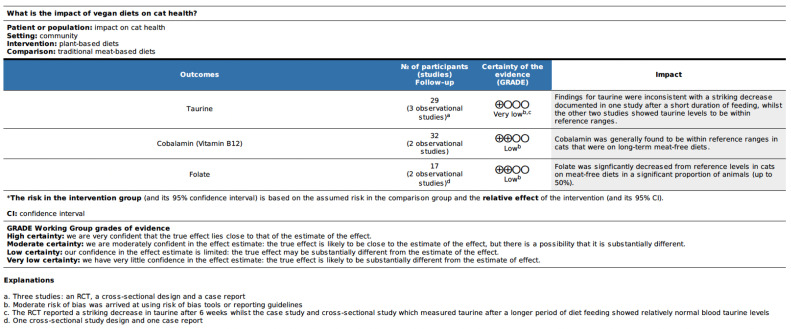
GRADE Summary of Findings table for animal-based outcomes of health assessed in cats fed a vegan diet.

**Figure 10 vetsci-10-00052-f010:**
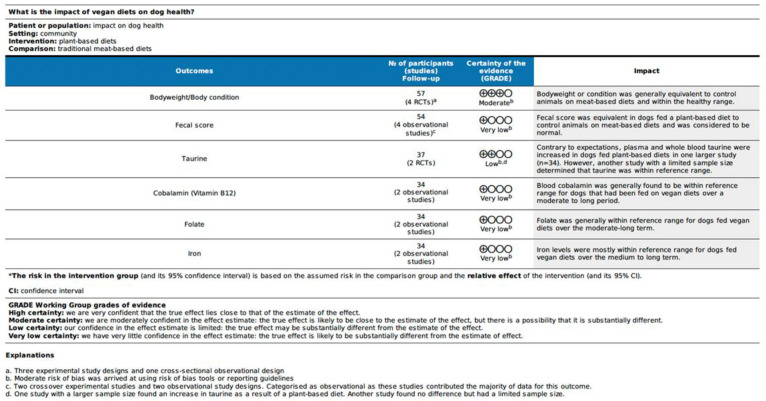
GRADE Summary of Findings table for animal—based outcomes of health assessed in dogs fed a vegan diet.

**Figure 11 vetsci-10-00052-f011:**
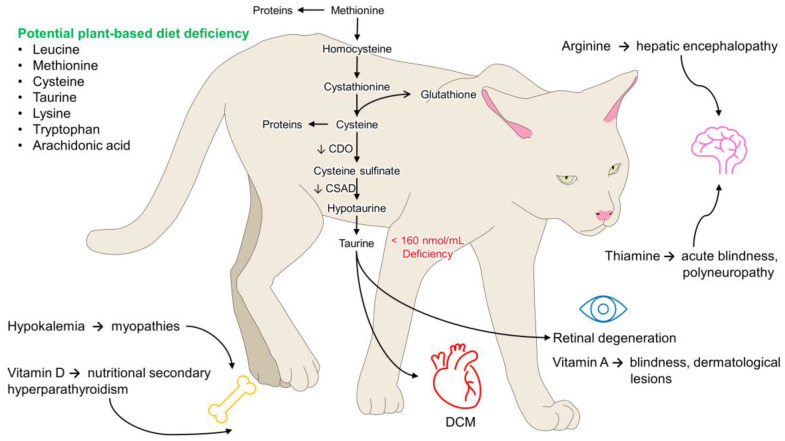
Potential adverse effects of nutritional deficits in vegan diets for domestic cats based on principles of nutritional physiology. Amino acid deficits are often reported in the analytical composition of vegan diets. Taurine, an amino acid required for correct cardiac and visual functioning, is essential to prevent retinal degeneration of DCM. Other deficiencies, such as arginine, vitamin D, vitamin A, or thiamine, can cause hepatic encephalopathy, secondary nutritional hyperparathyroidism, blindness, or polyneuropathies, respectively. DCM: dilated cardiomyopathy.

**Table 3 vetsci-10-00052-t003:** Direction of effect for health parameters in dogs fed vegan diets compared to meat-based diets.

Author	Comparator (If Applicable)	Outcome Assessed	Direction of Effect
Brown et al., 2009 [32]	MB	Food intake	↓
		Complete blood count	Within RR
Cavanaugh et al., 2021 [33]		Body weight	=
		BCS	=
		Essential AAs	↑
	MB	Taurine	↑
		Urine pH	↑
		Echocardiography	=
		TMOA	=
		Choline	↓
		Betaine	↓
		Creatinine	Within RR
El-Wahab et al., 2021 [35]	N/A	Food and water intake	=
		Body weight	=
		Fecal score	=
		Digestibility	=
Ingenpaß et al., 2021 [36]	MB	Fecal quality	=
		Apparent digestibility	=
Kiemer 2020 [37]	MB	Folic acid	Within RR
		Total protein	Within RR
		Iron	Within RR
		Vitamin B12	Within RR
		Calcium	Within RR
		Magnesium	Within RR
		Taurine	Within RR
		L-Carnitine	↑
		Physical examination	=
		Skin and coat	Healthy
		Stool consistency	=
Semp 2014 [27]	N/A	General appearance	Alert, responsive and playful
		Skin and coat	Inconspicuous, shiny and neat
		BCS	3–6/9
		Cardio system	Normal
		Respiratory tract	Normal
		Defecation	Normal
		Serum total protein	100% within RR
		Folic acid	71% within RR
		Vitamin B12	75% within RR
		Iron	90% within RR
		L- Carnitine	57% within RR
		Urinary pH	95% within RR
Rankovic et al., 2020 [38]	Four commercial foods	Glycemic index	=
		Glycemic response	=
		Insulinemic response	=
		BCS	=
Richards et al., 2021 [39]	Four commercial foods	Gastric emptying	↑

AA: amino acids; BCS: body condition score; IA: if applicable; MB: meat-based diet; N/A: not applicable; PBD: plant-based diet; RR: reference range; TD: traditional diet: TMOA: trimethylamine-N-oxide. ↑—increased with respect to comparator, ↓—decreased with respect to comparator =—equivalent to comparator.

**Table 4 vetsci-10-00052-t004:** Direction of effect of guardian-derived parameters in dogs fed vegan diets.

Author	Comparator (If Applicable)	Outcome Assessed	Direction of Effect
Dodd et al., 2022 [40]	MB	Health conditions	↓
		BCS	57% ideal
		Lifespan	↑
Knight and Satchel 2021 [26]	MB	Palatability	=
Knight et al., 2022 [41]	MB	Veterinary visits	↓
		Medication use	↓
		Perception of health	↑
		Health disorders	↓
Semp 2014 [27]	N/A	Perception of health	Normal or ↑
		Stool consistency	Normal
		Coat condition	Healthy

BCS: body condition score; MB: meat-based diet; N/A: not applicable.

## Data Availability

All data are reported within the text.

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
