# Peer review of "The Impact of Vegan Diets on Indicators of Health in Dogs and Cats: A Systematic Review"

_vetsci, 2023, doi:10.3390/vetsci10010052_

Round 1

Reviewer 1 Report

Well done on this important work. Please see attached.

Author Response

Reconsider the recommendation that homemade diets may be acceptable if commercially supplemented. In light of some studies showing that such diets are often nutritionally unsound (e.g. unbalanced), I suggest deleting this recommendation. I have 8 examples of such studies and have not done a recent search. E.g. Wilson et al 2019 ‘Evaluation of the nutritional adequacy of recipes for home-prepared maintenance diets for cats, Stockman et al 2013 ‘Evaluation of recipes of home-prepared maintenance diets for dogs’. If this recommendation is not deleted, then some brief discussion of such studies/concerns should be included, with a rationale for providing this recommendation in light of these. 

The other recommendation – use of nutritionally-sound commercial diets – should be retained. 

Thankyou- we do agree with this point and have deleted the recommendation to use commercial supplements as we did not review the literature on this topic. We have left in the recommendation to use nutritionally sound commercial diets. This has been deleted in the simple summary, abstract, discussion and conclusion.  

 The reviewed studies not only provide little evidence of adverse effects from plant-based diets, but also some evidence of benefits (notably in the surveys – these are included in your systematic review, but these benefits are little-mentioned). This should be acknowledged whenever this is mentioned (that there is ‘little evidence of adverse effects’, including simple summary, abstract, discussion, conclusions). This could be qualified, if you wish, by noting that much such evidence derives from reported outcomes/data in pet guardian surveys, which is not always reliable, but it should not be left out completely. 

We have added statements on benefits found from studies in the simple summary, abstract and conclusion as well as ensuring it is elsewhere in the body of the work.  

 The term ‘plant-based’ is used throughout. You discuss this under Methodology. Given that some plant material is used within most pet food, it seems vegan or vegetarian would be better terms. These accurately capture the ‘no meat’ meaning, and are also simple/clearly understood by readers (vs. explanations that plant-based means xx+% comes from plants, or similar). In fact, you note that you prefer to use the term ‘plant-based’ to denote a complete absence of animal products, but such diets are, in fact, vegan diets. Using ‘plant-based’ when you mean vegan is confusing. Please use ‘vegan’ throughout when you mean complete absence of animal products. That is correct, and also clear. If you use author terminology in places, note that you’re doing so (which you sometimes do), and that such author terminology has not always been consistent, and may or may not denote vegan.  

(nb: some critics state animals cannot be vegan as this is a wider lifestyle choice and they don't make such choices, but it remains valid to use the term ‘vegan diets’ for cats and dogs, and this term should be used when it’s accurate). 

Throughout the document we changed the term to “vegan” or specified that we were using the definition “plant-based diet” as in the original study. 

Likewise (this time just a recommendation), consider ‘guardian’ rather than ‘owner’ throughout, as some critique ‘owner’, partly as not all dogs and cats have a recognised owner (e.g. shelter animals, community owned animals). 

The terminology ‘owner’ has been changed to ‘guardian’ throughout.  

This field is undergoing rapid development with new studies being published. A key recent vegan dog health study is Davies (2022) (see https://sustainablepetfood.info/vegetarian-canine-diets/). This was not available at the time of your systematic review, but you should note in Discussion/Conclusions that ‘this field is undergoing rapid development with new studies regularly published’, and very briefly mention this more recent study, and key data from it: nos of animals, and very brief summary of findings. 

Thankyou for this reference. It is a shame that this missed coming up in our searches. We have added the paragraph below to highlight it and the increased research interest in the area 

“This field is attracting more attention by researchers as well as the pet food industry, with new studies regularly being published. In fact a recent study was sourced after the formal searches for this review.  Davies (2022) [49] evaluated the health perception of 100 guardians providing an extruded vegan food to dogs. The findings were overtly positive with The food being palatable, and leading to no adverse changes in appetite or body weight. Health improvements were also reported by the guardians in the following areas:  activity level (28%), fecal consistency (38.5%), antisocial smelling flatus (73.1%), coat glossiness (49.0%), itching (60.6%), skin redness 44.4%), and even in behavioral traits such as aggression (25.0%) or coprophagia (42.9%).It is hoped that within the next few years recommendations on this topic may be able to be made with a greater degree of certainty given this research interest.” 

The study by Knight et al (2021) on diet palatability (both dogs and cats), either (i) does not relate to health and should be excluded from this health-focused systematic review, or (ii) could be considered part of pet mental health: if maintained on diets pets don’t enjoy, chronic negative affective experiences and adverse mental health impacts could result. If choosing (ii) ie retaining this study in the systematic review, then there needs to be a very brief discussion along these lines – i.e. pet mental health is part of health overall, and could be adversely affected if diets are poorly palatable. This would then justify inclusion of this study.  

Thank you for your recommendation. We decided to retain this study by adding the mental health aspect of diets and companion animal welfare. This has been added in the results under both the cat and dog sections – lines 388 and 788-792.   

Simple Summary 

  1. 15 change ‘mostly meat-based diet (dogs)’ to ‘largely…’ 

Amended 

  • l. 20 change ‘little scientific study’ to ‘relatively little…’ or ‘limited…’

Amended to 'limited' 

  1. 22 change ‘small sample sizes and are’ to ‘small sample sizes or are’ (as the surveys are large) 

For clarity, we have elaborated on this and it now reads: “In addition, whilst there have been a number of survey studies with larger sample sizes, these types of studies can be subject to bias in terms of nature and disposition of the respondents towards alternative diets, or since answers may relate to subjective concepts such as body condition.” 

  1. 23-34 after ‘there is little evidence of adverse effects arising in dogs and cats fed plant-based diets’ add ‘and some evidence of benefits’ 

Added: "There is however some evidence of benefits particularly arising from owner’s perceptions of the diets. " 

Abstract 

  • l. 31 change ‘carnivorous species’ to ‘carnivorous or omnivorous species’ (as dogs are omnivorous – this is not currently indicated and needs to be indicated)

Amended 

  1. 39 delete comma 

Amended 

  1. 41 after ‘no overwhelming evidence arising from use of these diets’ add ‘and some evidence of benefits’ 

Amended as suggested.  

Methodology 

Under eligibility criteria, the meaning of the letter labels P, I, C, etc, are unclear – pls clarify or delete these.  

Added the meaning for the PICOS criteria characteristic in systematic reviews of intervention.  

  1. 105 clarify that English and Spanish were the languages spoken by the authors. 

Added. 

Text and Fig. 1. Were any studies also identified from checking reference lists of retrieved studies? Currently the text indicates this was done, but not whether any studies were sourced in this way. If so, these could be mentioned, and included as a 3rd (initial) column in Fig. 1. But this one is only a recommendation.  

No studies were sourced this way. A note to this effect has been added in the results.  

  1. 123 indicate who the 3rd reviewers were (initials if co-authors) 

This has been added (I.S.). 

‘Data synthesis’: discuss here, that you’ve included ‘direction of effect’ based on animal and owner-reported parameters, in Tables, for dogs, but not for cats. Briefly explain why not for cats.  

We believe we have reported the direction of effect for cats- this was however done narratively in the results section. Since there were fewer studies we did not think it worth an extra table for reporting this and we felt clarity was not an issue reporting this in text. We haven't amended the manuscript based on this suggestion. 

Results  

  • • Fig. 2 the x-axis should include ‘2021 – 2022’  

This change has been made.  

  • • l. 177 change to ‘a case report…’  

Amended 

  •  l. 204 ‘CK activity’ – define abbreviations when 1st encountered in the text. Not all readers will be vets – this article will attract a much wider readership.  

Thankyou, this has now been defined.   

  • • L. 229 ‘except 3 cats’ – those 3 were reportedly fed partly on table scraps (which are not nutritionally sound). This point would be worth making – if that info is in the study (I got it from the author I think). If not in the study, don’t bother. I suggest check the study.  

This was included in the text because it was included in the original study. 

  • L. 244 ‘Owner acceptance’ – their reports of health benefits ie clinical outcomes, are much more important than their acceptance, hence this section should be renamed e.g. to ‘Guardian reported health effects’. Ideally the text should be rewritten with the health effects 1st (e.g. the stat sig effects reported by Dodd et al 2021 – decr GI, hepatic and BW disorders, increased likelihood of +ve health assessment), followed by less important aspects.  

The title was changed and we moved the suggested lines to the beginning of the section. 

  • • L. 303 ‘TMAO’ – define abbreviations when 1st encountered in the text.  

This has been done throughout the manuscript.  

  • • L. 338: brackets needed  

      The brackets were included. 

  • • L. 356: typo  

              The typo was corrected. 

Tables  

  • • list the studies in chronological order: most recent to oldest, not alphabetic by author (as that’s not scientifically relevant, but how recent is – e.g. older diets are less likely to be manufactured to modern standards).  

 This is a pretty standard way of formatting details of included studies tables in Systematic reviews since it allows interested readers to just hone in on interesting studies easily if they don’t wish to read the whole study so we would prefer to keep this layout.  

  • • Add horizontal lines between studies to make reading easier  

The lines were added. 

  • • Consider not just listing an overall ‘N’ but also adding N meat-based, N vegan (or plant-based, etc, depending on author terminology). This would be a very useful/interesting extra piece of information – strongly recommended.   

We believe this was done whenever the authors made this information available. Where it is not within tables it is generally because there was no comparator (usually the meat-based control is missing) or the study design is a crossover, hence n is the same for both arms as listed in the diet type column.  

  • Table captions should normally be above tables (unless different for this journal?) – check all Tables  

Captions for Table 4 and 5 were corrected. 

Cat studies  

  • Where 1+ authors exist, add ‘et al’ as in Dodd et al. 2021.  

Amended throughout the manuscript. 

  • • Check the Dodd et al. 2021 cat study. I think it included including dietary information for 1,026 cats, of whom 187 were fed vegan diets. Hence the sample size should be 1,026. I think the remainder did not provide this dietary info. Also, I think the lifespan, was for previously-owned cats. That would be a different no. So clarify in the Table as in ‘Lifespan (xx previously owned cats)’  

The sample size was included and the lifespan for previously-owned cats was specified. 

  • • The Knight et al 2021 study included a variety of normal domestic breeds, so many mixed breeds and some pure breeds. So state ‘various, unspecified’ or similar. (This point also applies to the Semp 2014 study). Also, of the cats in the Knight et al 2021 study, only 1,135 were included (guardians supplied sufficient info). Hence this should be the sample size. However, this study only reported palatability based on feeding behaviour – not health outcomes. Hence should it even be in this sys review focused on health outcomes? 

The changes for Knight and Semp studies were amended as per the suggestion.  The suggestion about associating palatability to mental health was incorporated to justify the inclusion of this study in the present systematic review. 

  • Knight et al 2021 and Semp 2014 column 1 have ‘study type’ at bottom of cell – delete (or clarify)  

This has been deleted.  

  • • Wakefield et al. 2006 column 3. Should be ‘Meat-based’. Column 4 should be ‘[diet A] or [diet B]’. Check this study. The owners also reported their perceptions of health status. This should be added to ‘body condition’  

We checked the Wakefield study and believe the table is correct. The 17 cats refers to the subset of the vegan groups that were blood sampled and blood sampling was not performed on a meat-based control as a comparator. The survey data did relate to animals in both groups.  

Dog studies  

  • • Cavanaugh 2021 and others: cannot read diet type as seems to run into the study above? Hopefully these will resolve, with addition of horizontal lines between studies to make reading easier  

The horizontal lines were added. 

  • • Kiemer 2020: feeding the diet for 3+ mths seems to conflict with feeding for 6 wks – clarify/resolve  
    The wording was improved since the trial was for six weeks and included dogs that have been consuming a vegan diet for at least three months. 
  • • Dodd et al 2022. The lifespan was for previously-owned dogs. That would be a different no. So clarify in the Table as in ‘Lifespan (xx previously owned dogs)’  

This has been clarified. 

  • • Knight et al 2021: as with the cat study, this study only reported palatability based on feeding behaviour – not health outcomes. Hence should it even be in this sys review focused on health outcomes?  

The suggestion about associating palatability to mental health was incorporated. A paragraph was added to justify the inclusion of this study in the systema review. 

  • • Knight at al 2022: there were 7 general health indicators (not just 4) and 22 specific health disorders assessed. The 7 should be listed and 22 specific health disorders mentioned (not all listed). Alternatively simply mention there were 7 general health indicators and 22 specific health disorders assessed.  

In the text, it was specified that 7 general health indicator and 22 specific health disorders were assessed. 

Table 3  

  •  Title ‘Direction of effect of animal-based parameters in dogs fed plant-based diets.’ Is a bit confusing. Clarify as in ‘Direction of effect on health parameters in dogs fed plant-based (or vegan) diets compared to meat-based diets’ (or similar)  

Amended as suggested.  

  • • Knight at el 2021: only reports palatability, should it be included?  

Your suggestion about associating palatability to mental health was incorporated 

Table 4. (GRADE summary – cats).  

  • • Folate impact – grammatical errors (… at in a significant…)  

The GRADE tables have been checked for grammatical errors and amended as necessary. They have also been amended to read ‘vegan’ rather than plant-based.  

Figure 9  

  • • ‘Plant-based diet deficiency: Leucine, Methionine, …’: change to ‘Potential plant-based diet deficiency…’  
  • • Delete the urolithiasis image and text. This earlier concern based on cases described in popular books/online, has not since been supported by large-scale studies (Dodd et al 2021, Knight et al forthcoming). This earlier concern appears to have been unfounded.  

Alternatively, delete this Figure. 

We made the suggested changes inside the figure. 

Discussion  

Owner perceptions  

  • • This section should be expanded. They key findings of the key studies in this section are presently little mentioned. They should specifically noted (wording variations are fine but the key facts should be included):  

Thankyou for this suggestion. We would prefer that this kind of langauge is in the results as opposed to the discussion since we think the latter should provide more of an overview of the results and bring it together. For this reason we have added your suggested changes but put them into the relevant (dog v cat) sections in the results. A couple of paragraphs have also been added to the discussion at lines s975-100.  

The suggested paragraph was included.  

  • • L. 476-478 ‘there is also evidence of interest in converting pets to a plant-based diets with 35% of surveyed owners who did not feed a plant-based diet expressing interest should certain conditions be met’.   

Add a summary of Knight et al (2022) study above “… the 1,370 respondents who used a conventional meat formulation as their dog’s normal diet, and the 830 who used a raw meat formulation. These combined 2,200 respondents were asked whether they would realistically choose alternative diets, if these offered their desired attributes. The alternatives offered for consideration were vegetarian and vegan diets, as well as those based on laboratory grown meat, insects, fungi and algae. Of 2,181 who answered this question, 44% (955) confirmed they would realistically choose such alternative diets.” [if certain conditions were met].  

The suggested paragraph was included at lines 1015-1023. 

  • • L. 499 change to ‘… and sometimes, small sample sizes’ (as the surveys are not small) 

This has been changed as suggested.   

Conclusions  

  • • l. 518 change to ‘often use low sample sizes or short feeding durations’ (as the surveys are not small)  

We changed this sentence as suggested.  

Reviewer 2 Report

The systematic review entitled “The impact of plant-based diets on indicators of health in dogs and cats: a systematic reviewapproaches a trending subject related to animals’ humanization by adoptind a plant-based diet that lacks evidence-based research. The use of the GRADE (Grading of Recommendations, Assessment, Development and Evaluation) methodology enhanced the review’s quality and reliability.

Overall, the review is well structured following PRISMA guidelines, and the sections are well organized. The results are properly interpreted, and the recommendations are relevant.

However, there are certain revisions to consider:

·         Line 47: the last two keywords “vegan companion” and “animal diets” should be divided by a semicolon.

·         Line 195: Please rectify the word “considered”.

·         Line 197: Please rectify “e.g.”.

·         Line 204 and 303: Please write the full words when first used.

·         Line 377: Please rectify the word “committee”

·         Table 1: The table lacks a footnote.

The manuscript discusses appropriately certain limitations of plant-based diets on cats and dogs such as taurine deficiencies which has been mentionned. However, there is no further discussion on the nature of these plant-based diets (plants species) which is an important factor to take into consideration.

Author Response

Line 47: the last two keywords “vegan companion” and “animal diets” should be divided by a semicolon.

Amended

Line 195: Please rectify the word “considered”.

Amended

Line 197: Please rectify “e.g.”.

Amended

Line 204 and 303: Please write the full words when first used.

Thankyou- added creating kinase and trimethylamine-N-oxide in text at these points. 

Line 377: Please rectify the word “committee”

Amended

Table 1: The table lacks a footnote.

It is not clear that this is needed since there are few abbreviations within this table. A short one has been added for the description of males/females. 

The manuscript discusses appropriately certain limitations of plant-based diets on cats and dogs such as taurine deficiencies which has been mentionned. However, there is no further discussion on the nature of these plant-based diets (plants species) which is an important factor to take into consideration

Thankyou for this suggestion. We have now added a paragraph in the introduction at lines 68-78 elaborating on the issues with nutrient deficiencies in plant based diets and describing the common protein sources used. This paragraph reads:

The dominant proteins in plant-based pet foods have historically been soy, corn protein and wheat protein (gluten). Recently additional plant proteins have become available for use, including pea protein, potato protein and rice protein, and based on trends in human nutrition this may expand further to include others such as hemp, oat and bean proteins [13]. In contrast to animal tissues, plant cells are rich in carbohydrates (e.g., cellulose) that carnivores have difficulty digesting [11]. Proteins from cereal grains or soy contain lower amounts of essential amino acids. These include sulfur amino acids and the omega-3 fatty acids eicosapentaenoic acid and docosohexaenoic acid. Typically they do not contain all essential vitamins, e.g. retinol (Vitamin A) and cobalamin (vitamin B12) [12]. Additionally, plants may contain toxic compounds that only the gastrointestinal tract of herbivores can detoxify [11].